# Risk variants of obesity associated genes demonstrate BMI raising effect in a large cohort

Muhammad Saqlain[1,2], Madiha Khalid[1], Muhammad Fiaz[3], Sadia Saeed[1], Asad Mehmood Raja[1], Muhammad Mobeen Zafar[1], Tahzeeb Fatima[4], João Bosco Pesquero[5], Cristina Maglio[4,6,7], Hadi Valadi[4], Muhammad Nawaz[4]*, Ghazala Kaukab Raja[1]*

1 University Institute of Biochemistry & Biotechnology, PMAS- Arid Agriculture University Rawalpindi, Rawalpindi, Pakistan, 2 Department of Biochemistry and Molecular Biology, University of Sialkot, Sialkot, Pakistan, 3 Department of Pathology, Pakistan Institute of Medical Sciences (PIMS), Islamabad, Pakistan, 4 Department of Rheumatology and Inflammation Research, Institute of Medicine, Sahlgrenska Academy, University of Gothenburg, Gothenburg, Sweden, 5 Center for Research and Molecular Diagnostic of Genetic Diseases–Department of Biophysics, Federal University of São Paulo, São Paulo, Brazil, 6 Region Västra Götaland, Sahlgrenska University Hospital, Rheumatology Clinic, Gothenburg, Sweden, 7 Wallenberg Center for Molecular and Translational Medicine, University of Gothenburg, Gothenburg, Sweden

* muhammad.nawaz@gu.se (MN); ghazala@uaar.edu.pk (GKR)

**Data Availability Statement:** All relevant data are within the paper and its supporting information files.

## Abstract

Obesity is highly polygenic disease where several genetic variants have been reportedly associated with obesity in different ethnicities of the world. In the current study, we identified the obesity risk or protective association and BMI raising effect of the minor allele of adiponectin, C1Q and collagen domain containing (*ADIPOQ*), cholesteryl ester transfer protein (*CEPT*), FTO alpha-ketoglutarate dependent dioxygenase (*FTO*), leptin (*LEP*), and leptin receptor (*LEPR*) genes in a large cohort stratified into four BMI-based body weight categories i.e., normal weight, lean, over-weight, and obese. Based on selected candidate genetic markers, the genotyping of all study subjects was performed by PCR assays, and genotypes and allele frequencies were calculated. The minor allele frequencies (MAFs) of all genetic markers were computed for total and BMI-based body weight categories and compared with MAFs of global and South Asian (SAS) populations. Genetic associations of variants with obesity risk were calculated and BMI raising effect per copy of the minor allele were estimated. The genetic variants with higher MAFs in obese BMI group were; rs2241766 (G = 0.43), rs17817449 (G = 0.54), rs9939609 (A = 0.51), rs1421085 (C = 0.53), rs1558902 (A = 0.63), and rs1137101 (G = 0.64) respectively. All these variants were significantly associated with obesity (OR = 1.03–4.42) and showed a high BMI raising effect (β = 0.239–0.31 Kg/m$^2$) per copy of the risk allele. In contrast, the MAFs of three variants were higher in lean-normal BMI groups; rs3764261 A = 0.38, rs9941349 T = 0.43, and rs7799039 G = 0.40–0.43). These variants showed obesity protective associations (OR = 0.68–0.76), and a BMI lowering effect per copy of the protective allele (β = -0.103–0.155 Kg/m$^2$). The rs3764261 variant also showed significant and positive association with lean body mass (OR = 2.38, CI = 1.30–4.34). Overall, we report six genetic variants of *ADIPOQ*, *FTO* and *LEPR* genes as obesity-risk markers and a *CETP* gene variant as lean mass/obesity protective marker in studied Pakistani cohort.

**Funding:** This research was funded by grant 20-2255/NRPU/R&D/HEC from Higher Education Commission (HEC) Pakistan to GK Raja. The funders had no role in study design, data collection and analysis, decision to publish, or preparation of the manuscript. Nawaz M acknowledges the support from Adlerbertska Research Foundation (2019).

**Competing interests:** The authors have declared that no competing interests exist.

**Abbreviations:** ADIPOQ, Adiponectin, C1Q and collagen domain containing; BMI, Body mass index; CDC, Centers for Disease Control and Prevention; CETP, Cholesteryl ester transfer protein; CHD, Coronary heart diseases; DBP, Diastolic blood pressure; FBS, Fasting blood sugar; FTO, FTO alpha-Ketoglutarate dependent dioxygenase; HDL-C, High density lipoprotein cholesterol; LDL-C, Low density lipoprotein cholesterol; LEP, Leptin; LEPR, Leptin Receptor; MAFs, Minor allele frequencies; NIH, National Institute of Health; ORs, Odds' ratios; RFLP, Restriction fragment length polymorphism; SAS, South Asian population; SBP, Systolic blood pressure; SNP, Single nucleotide polymorphisms; T2D, Type 2 diabetes; TC, Total Cholesterol; vLDL-C, Very low-density lipoprotein cholesterol; WHO, World Health Organization; TG, Triglycerides.

## Introduction

Obesity is a rapidly growing health risk worldwide. Overweight and obesity represent abnormally elevated ranges of weight, generally measured in terms of body mass index (BMI). Based on BMI, the body weight is categorized into lean body mass (BMI<18.5 Kg/m$^2$), normal weight (BMI 18.5–24.9 Kg/m$^2$), overweight (BMI 25–29.9Kg/m$^2$), and obese (BMI≥30 Kg/m$^2$) [1–3]. Out of range BMI values have been found to be associated with certain health problems, especially obesity that has been declared as the epidemic of this century [4, 5]. Since 1980s, the incidence of obesity has increased up to three folds in many countries with an alarming increase in the number of affected individuals [6]. According to WHO 2016 data, there are almost 2 billion overweight and 650 million obese adults globally. If similar rates continue, it is expected that the overweight population would reach 2.7 billion and obese individuals above 1 billion by 2025 [7]. This obesity epidemic has also led to a global rise in many life-threatening diseases, especially coronary heart diseases (CHD), type 2 diabetes (T2D), hypertension, among others [8]. Major factors contributing to rise in obesity are lifestyle, environmental, as well as genetic predisposition [9–14].

Common phenotypes of overweight and obesity are polygenic, heritable, and multifactorial, where most of the susceptibility to obesity is related to common DNA variants [15]. Several studies have reported BMI-associated candidate genes, and related genetic polymorphisms as well as their interactions with obesogenic environmental inductions in different populations [9, 10, 12, 16, 17]. According to 2014 statistics by WHO, Pakistan was declared the 9$^{th}$ most obese country in the world with almost 5.4% adult obese individuals (>30kg/m$^2$) and 23% overweight (>25kg/m$^2$) [18, 19]. The main contributing factors are less physical activity and unhealthy eating habits, which lead to increased prevalence of obesity in Pakistan [2]. In this perspective, it is important to explore a genetic panel representing the common variants influencing BMI in this region. The research in the field of genetic markers influencing BMI phenotypes in the Pakistani populationhas been very limited [2]. As 1000 genomes data indicates (http://www.internationalgenome.org), the allele frequencies of genetic variants widely differ among different world populations and ethnicities. The ethnicity-specific frequency distribution of these variants also tends to vary in individuals of a population/ethnic background based on their (risk associated/or protective) susceptibility towards a disease which may be due to their exposure to population-specific natural selection [9, 10, 20, 21].

Previously, it has been reported that the genetic variants in *ADIPOQ*, *CETP*, *FTO*, *LEP*, and *LEPR* genes are strongly correlated with obesity and associated metabolic complications in different ethnicities of the world [9, 10, 22–30]. Based on the potential obesogenic roles of *ADIPOQ*, *CETP*, *FTO*, *LEP*, and *LEPR* genes single nucleotide polymorphisms (SNPs), the aim of the current study was to identify obesity risk markers through MAFs estimation, genetic association and BMI raising risk effect analysis using a set of candidate genetic variants in a BMI stratified Pakistani cohort. Despite being at high risk of obesity, the Pakistani population lacks prior studies on genetic susceptibility to weight gain and genotype distribution data of potential obesity predisposing genetic variants [2]. The findings of the current study could be useful in setting up a gene panel to identify individuals highly prone to weight gain as well as individualized weight management plans.

## Methods

The study cohort consisted of 4000 subjects further grouped into four BMI based lean, normal weight, overweight, and obese categories of 1000 individuals each. Blood samples were collected from outdoor patient department (OPD) of local hospitals in Rawalpindi and Islamabad, Pakistan. The ethical approval for the study was obtained from the Ethics Committee for

the Use of Human Subjects, Pir Mehr Ali Shah Arid Agriculture University Rawalpindi (PMA-S-AAUR) and Pakistan Institute of Medical Sciences (PIMS), Islamabad, Pakistan. The informed written consent was signed by the subjects (participants). After getting the signed consent the data was collected on a performa by interviewing the subjects (participants). The subjects who refused to take part in the study were excluded. Based on self-reported ethnicity, the majority of the study subjects were Punjabi with a smaller distribution of Pashtuns and Kashmiris. Anthropometric and biochemical examinations were performed in the hospital (PIMS) laboratory. This includes the blood pressure (systolic blood pressure; SBP and diastolic blood pressure; DBP), fasting blood sugar (FBS, levels of ≤100mg/dL were used as reference), triglycerides (TG), total cholesterol (TC), high density lipoprotein cholesterol (HDL-C), low density lipoprotein cholesterol (LDL-C), and very low density lipoprotein cholesterol (vLDL-C) as described previously [8]. A ratio of body weight (in Kg) to the height (in meter square) was computed to estimate the BMI ($kg/m^2$). The subjects were categorized based on the WHO defined obesity categories; lean (<18.5 $Kg/m^2$), normal weight (18.5–24.9 $Kg/m^2$), over-weight (25.0–29.9$Kg/m^2$), obese (≥30 $Kg/m^2$). The subjects who were pregnant or diagnosed with terminal health illnesses such as cancer, viral hepatitis, and advanced stage liver disorder(s) were not included in the study.

SNPs of five candidate genes, *LEPR*, *LEP*, *ADIPOQ*, *FTO*, and *CETP*, were selected based on their involvement in different energy balance mechanisms and having significant association with BMI and obesity [22–27]. For genotyping, one SNP each from *LEPR*, *LEP*, *ADIPOQ*, and *CETP* genes were selected, whereas six SNPs of *FTO* gene were selected due to their widely known association with BMI and obesity.

## Isolation of genomic DNA

Genomic DNA was extracted using organic solvent extraction method [31]. DNA pellet was dissolved in Tris EDTA (TE) buffer and small aliquots of DNA stocks were stored at -20˚C. Quantification of isolated DNA was performed by NanoDrop (Cuvdrop Avan, Taiwan) micro-volume spectrophotometer. The integrity of extracted DNA was confirmed on 1% agarose gel [31].

## PCR based genotyping

For PCR based genotyping assays, amplification refractory mutation system-polymerase chain Reaction (Tetra-ARMS PCR) and Allele-Specific Oligonucleotide PCR (ASO-PCR) primers were designed using BATCH Primer3 and IDT SciTools [32, 33]. For *LEP* and *LEPR* SNPs, restriction fragment length polymorphism (RFLP) assay was designed using web based tool NEBcutter V2.0 for the selection of restriction enzyme and expected amplified cut/un-cut SNP product size [34]. All PCR products and restriction fragments were analyzed by agarose gel electrophoresis. The list of primers used in this study are provided in S1 File.

## Statistical analysis

Means and standard deviation (Mean±SD) were computed for all descriptive characteristics. The allele and genotype frequency distribution were calculated in overall and BMI-based sub-categories (n = 3827). The deviation of the genotypes of all studied genetic variants from Hardy–Weinberg equilibrium (HWE) were tested using Chi-square test ($\chi^2$). The descriptive data was analyzed using SPSS version 21.0 [35].

The associations of genetic variants with obesity and other BMI based categories were computed using logistic regression. Odds' ratios (ORs) with 95% confidence intervals (95%CI) were calculated while adjusting for age and gender to overcome the bias arising from these

confounding variables. Genetic association analysis of SNPs was carried out in lean vs over-weight-obese combined BMI based groups.

The BMI raising effect size of the minor allele of all genetic variants was measured by calculating β-coefficients (SE) using linear regression analysis. The β-coefficients (SE) represent the change in BMI ($Kg/m^2$) for a copy of the minor allele of a variant carried by an individual in total population as well as in males and females to test sex-specific risk effects. Genotype/allele frequencies and association analysis for all SNPs were calculated using PLINK and SNPStats online tools [36]. To account for any error resulting from multiple SNPs tested on each sample, Bonferroni adjusted p-values (p-value≤0.005) were used for genetic association results. For all other parameters statistical significance was considered as p-value ≤ 0.05.

## Results

### Characteristics of population cohort

The descriptive characteristics of population cohort are presented in **Table 1**. As presented in **Table 1** the mean levels of all studied parameters (Weight, BMI, WC, SBP, DBP, FBS, TG, TC, HDL, LDL, and vLDL) were within normal ranges along with below and above normal ranges predicting the health consequences for over-weight and obese subjects.

### Minor allele frequencies are highly raised in total and all BMI based sub-groups of studied Pakistani cohort

The MAFs of all genetic variants in total studied population, global and SAS populations are presented in **Table 2**. The MAFs of *ADIPOQ* rs2241766 G (0.36), *CETP* A rs3764261 (0.35), *FTO* rs17817449 G (0.43), rs9939609 A (0.47), rs1421085 C (0.42), rs1558902 A (0.46), rs9941349 T (0.41), and *LEP* rs7799039 A (0.66) were higher in total studied population compared to SAS and global populations (**Table 2**).

Two SNPs with MAFs lower than the global and SAS populations were, *FTO* rs7204609 C (0.04) and *LEPR* rs1137101 A (0.43). In case of *LEP* rs7799039 SNP, the G allele was considered

**Table 1. Descriptive statistics of study population.**

| Study Variables | Minimum | Maximum | Mean±SD |
|---|---|---|---|
| **Age (Years)** | 20 | 60 | 35.51±10.98 |
| **Body Weight (Kg)** | 19 | 145 | 71.19±18.75 |
| **Height (cm)** | 125 | 210 | 166.14±11.89 |
| **BMI ($Kg/m^2$)** | 16 | 42.97 | 25.05±7.31 |
| **WC (cm)** | 25.4 | 220.90 | 98.85±30.95 |
| **SBP (mmHg)** | 81 | 210 | 133.49±19.34 |
| **DBP (mmHg)** | 60 | 130 | 82.34±10.88 |
| **TC (mmol/L)** | 1.07 | 61.36 | 4.48±1.86 |
| **HDL (mmol/L)** | 0.014 | 3.6 | 1.17±0.45 |
| **LDL (mmol/L)** | 0.09 | 61.07 | 2.97±1.85 |
| **TG (mmol/L)** | 0.3 | 9.77 | 1.73±1.02 |
| **vLDL (mmol/L)** | 0.14 | 1.95 | 0.3±0.20 |
| **FBS (mmol/L)** | 2.3 | 35.944 | 5.90±2.25 |

WC; waist circumference, SBP; systolic blood pressure, DBP; diastolic blood pressure, TC; Total Cholesterol, HDL; high density lipoprotein, LDL; low density lipoprotein, TG; triglycerides, vLDL; very low-density lipoprotein, FBS; fasting blood sugar. FBS levels of ≤100mg/dL were used as reference.

**Table 2. Details of genetic variants and MAFs in global, SAS and studied population.**

| Genes | Chromosomal Location | SNPs | Major/ Minor Allele | Global MAF | SAS MAF | Studied MAF (Total) |
|---|---|---|---|---|---|---|
| *ADIPOQ* | 3q27 | rs2241766 | T/G | 0.1514 | 0.151 | 0.36 |
| *CETP* | 16q21 | rs3764261 | C/A | 0.2895 | 0.321 | 0.35 |
| *FTO* | 16q12.2 | rs17817449 | T/G | 0.3077 | 0.289 | 0.43 |
| | | rs9939609 | T/A | 0.3401 | 0.288 | 0.47 |
| | | rs1421085 | T/C | 0.2286 | 0.307 | 0.42 |
| | | rs1558902 | T/A | 0.2280 | 0.307 | 0.46 |
| | | rs9941349 | C/T | 0.2710 | 0.367 | 0.41 |
| | | rs7204609 | T/C | 0.2181 | 0.072 | 0.04 |
| *LEP* | 7q31.3 | rs7799039 | G/A | 0.4016 | 0.514 | 0.66 |
| *LEPR* | 1p31 | rs1137101 | A/G | 0.5843 | 0.503 | 0.43 |

(Data Source: 1000 Genomes Project Phase 3). MAF; Minor Allele Frequency, SAS; South Asian population

as minor allele for further analysis due to very high frequency of A allele in current population cohort. The BMI categories (Lean, over-weight, obese and over-weight-obese combined vs. normal) based comparison of the MAFs is shown in **Fig 1**. The SNPs having higher MAFs in over-weight and obese compared to the lean and normal BMI include; *FTO* rs17817449 G (0.54–0.59 vs 0.49), rs9939609 A (0.51 vs 0.39), rs1421085 C (0.42–0.53 vs 0.38), rs1558902 A

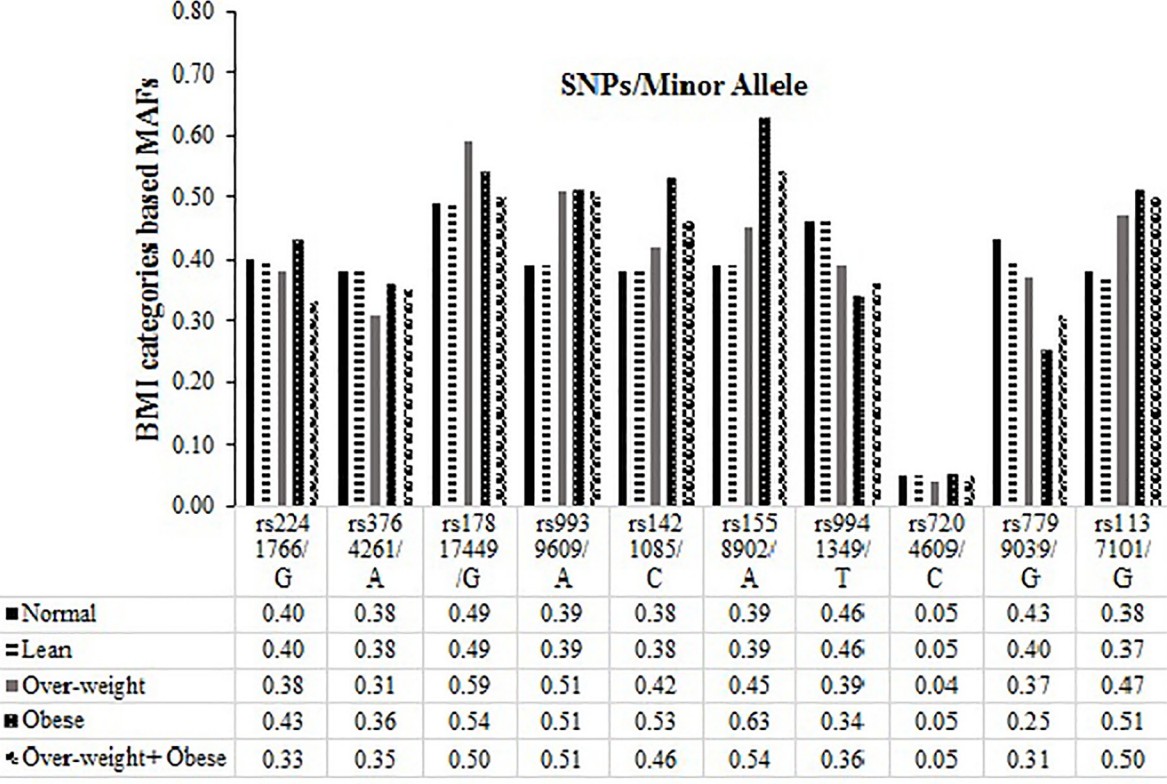

**Fig 1. Comparison of minor allele frequencies (MAFs) based on BMI categories.** The BMI categories-based distribution of the MAFs of all genetic variants is presented as bar graph. The y-axis represents frequencies of the minor alleles per BMI categories (Normal, Lean, Over-weight, Obese, and Over-weight + Obese) while genetic variants with corresponding minor alleles are shown in the x-axis. MAFs were calculated using online software SNPStats.

(0.45–0.63 vs 0.39), and *LEPR* rs1137101 A (0.47–0.51). Whereas *ADIPOQ* SNP rs2241766 had slightly higher MAF (G = 0.43) in obese category compared to normal and lean (0.40). Genetic variants with higher MAFs in lean-normal BMI categories compared to the over-weight and obese were; *CETP* A rs3764261 (0.38 vs 0.31–0.35), *FTO* rs9941349 T (0.46 vs 0.34–0.39), and *LEP* rs7799039 (0.40–0.43 vs 0.25–0.37), Whereas *FTO* rs7204609/C frequency was within similar ranges (0.05) among all BMI groups (**Fig 1**). The genotypes of all studied SNPs were found to be within Hardy–Weinberg equilibrium (p>0.05).

## Association analysis of genetic variants based on BMI categories

Keeping with main objective of our study, genetic association analysis was performed using BMI only BMI. The genetic association results of studied SNPs with BMI based categories of lean and overweight-obese (combined) verses normal body weight (as a reference group) are presented in **Fig 2**. Genetic variants which showed strong risk associations with overweight-obese BMI category were; rs1137101 (OR = 4.42, CI = 2.76–7.08, p = $2\times10^{-6}$), rs1558902 (OR = 3.03, CI = 5.8–8.4, p = $4\times10^{-5}$), and rs2241766 (OR = 2.50, CI = 3.1–5.40, p = $3\times10^{-6}$) whereas three *FTO* variants; rs9939609, rs1421085, and rs17817449 showed statistically significant yet mild risk associations (OR = 1.03–1.44, p = $1\times10^{-6}$). Overall, the minor alleles of all above variants showed increasing risk of obesity ranging from 4.4 folds to 1.03 folds. In contrast, only rs3764261 variant showed statistically significant association with lean body mass (OR = 2.38, 95% CI = 1.30–4.34, p = $1\times10^{-6}$), indicating the minor allele carriers are 2.38 times likely to have lean body mass. Three genetic variants having protective associations were, rs3764261 (OR = 0.76, CI = 0.68–0.85, p = $9\times10^{-7}$), rs9941349 (OR = 0.68, CI = 0.61–0.77, p = $6\times10^{-5}$) and rs7799039 (OR = 0.69, CI = 0.63–0.76, p = $8\times10^{-5}$) indicating a 76%-68% lesser risk of obesity respectively. One genetic variant, rs7204609, lacked associations with any of the BMI based obesity category (**Fig 2**).

## BMI raising risk

The MAFs and association analysis results of genetic variants were further replicated to examine the BMI raising risk effect of the minor alleles of studied genetic variants (**Fig 3**). The BMI raising effect of the minor alleles of all genetic variants was measured as β coefficients (SE)

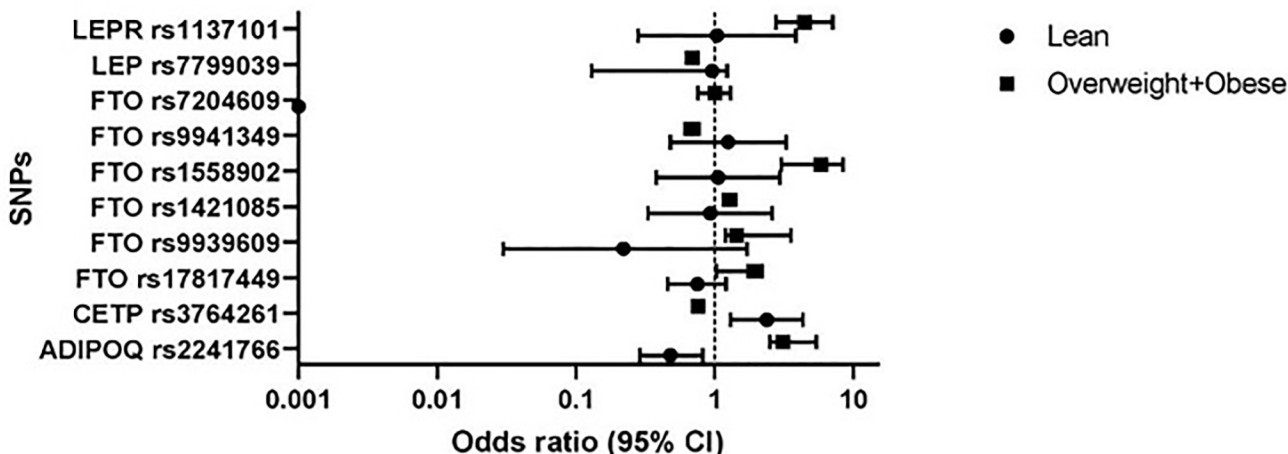

**Fig 2. Association analysis of minor alleles with BMI stratified categories.** Associations of genetic variants with BMI based categories (Lean and Over-weight + Obese) were obtained by Odds ratios (ORs) with 95% confidence intervals (95% CI). ORs (95% CI) were calculated using age and gender adjusted logistic regression models and normal BMI category considered as reference group with an OR = 1.0.

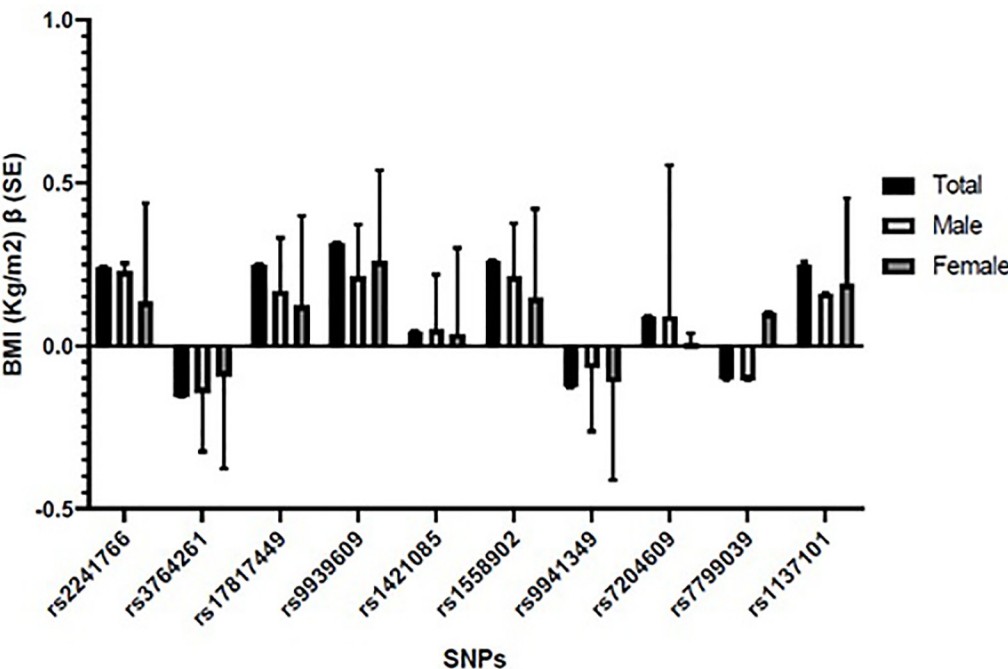

**Fig 3. BMI raising risk effect size estimation of minor alleles.** The BMI raising risk effect size of all genetic variants are presented as β-coefficients (SE) in total and gender based (Male vs female) population cohort obtained by linear regression analysis. The positive β-coefficient values represent an increase in BMI (Kg/m²) while a negative value showed a decrease in BMI (Kg/m²) for each copy of the minor allele of a variant.

representing an increase in BMI (Kg/m²) per copy of the minor allele of a variant carried by an individual. A comparison of the β coefficients for total studied population, males and females is shown in Fig 3. The genetic variants and minor/risk alleles showing positive/BMI raising effect sizes are in following descending order; rs9939609 A (β = 0.314Kg/m², SE = 0.002) followed by rs17817449 G (β = 0.249Kg/m², SE = 0.002), rs1137101 G (β = 0.285Kg/m², SE = 0.002), and rs2241766 G (β = 0.239Kg/m², SE = 0.004) alleles (**Fig 3**). Overall, our results show that the above variants lead to 0.314–0.285Kg/m² higher BMI in the presence of their single risk allele. In contrast, the minor A allele of rs3764261 (β = -0.155Kg/m², SE = 0.001), T allele of rs9941349 (β = -0.125Kg/m², SE = 0.001) and G of rs7799039 (β = -0.103Kg/m², SE = 0.002) showed negative effect sizes thus indicating that their minor alleles have an obesity protective effect. A single copy of their minor alleles leads to -0.155 to -0.103Kg/m² lower BMI.

Except a few sex specific trends, the BMI raising risk allele effect sizes in males and females were in similar ranges to that found in total population (**Fig 3**). The rs9939609 and rs1137101 SNPs had higher effect sizes in females (β = 0.259, p = 4.7x10⁻¹¹ and 0.294 Kg/m², p = 7.5x10⁻²⁰) as compared to males (β = 0.211, 0.281 Kg/m²), thus showing their minor allele carrier females tend to have 0.048–0.013Kg/m² higher BMI than their male counterparts. Two variants, rs3764261 and rs7799039 showed sex specific negative effect sizes (**Fig 3**) indicating an obesity protective role of their minor alleles. However, the negative effect size of rs3764261 variant was higher in males, vs females (β = -0.142, vs -0.093Kg/m², p = 1.8x10⁻¹²) and of rs7799039 higher in females as compared to males (β = -0.294 vs -0.281 Kg/m², p = 5.43x10⁻⁸).

## Discussion

In current study, we report the associations of six genetic variants with obesity risk, three with obesity protection and one with lean body mass in a cohort of individuals from Pakistan. The

minor allele frequencies of all variants were compared between BMI based body mass groups, lean-to- obese. The MAFs of the six variants of the *LEPR*, *ADIPOQ*, and *FTO* genes were higher in obese individuals, and they also showed strong association with over-weight/obese BMI (excess body weight). In case of the three genetic variants of *CEPT*, *FTO* and *LEP* genes, MAFs were higher in lean-normal body mass and showed obesity protective association. Of all studied markers, only CEPT genetic variant showed significant association with lean body mass.

Obesity is on rise globally and there is need of identification of genetic markers with best capturing maximum disease effects. The polygenic nature of human obesity, strong gene-gene, and gene to environment interactions, and above all large inter-individual and ethnic variations are potential hurdles in the discovery of accurate genetic marker(s) for human obesity [2, 37]. Three genes that have gained much attention are; *LEP*, *LEPR*, and *FTO* [9, 10, 38–40]. The obesogenic roles of these genes have been documented through familial, twin and population-based studies [2, 41, 42]. Of several variants in *LEP*, *LEPR*, and *FTO* genes identified and tested for obesity association, only a few have been successfully replicated in different world populations [37]. The common disease markers in these genes have been reported to increase susceptibility towards obesity either directly or in combination with other gene(s)/variant(s) [37–40].

In the present study, we report associations of ten SNPs from five candidate genes with BMI in a large Pakistani cohort. The MAFs of all genetic variants showed marked variations in total studied population, BMI based lean-obese subgroups (**Fig 1**), global and SAS populations (**Table 2**). The MAFs of all studied genetic markers were raised in obese BMI study group (*ADIPOQ* rs2241766 G (0.43), *FTO* SNPs; rs17817449 G (0.54), rs9939609 A (0.51), rs1421085 C (0.53), rs1558902 A (0.63), and *LEPR*rs1137101 (0.51). All genotyped markers showed highly significant associations ($p < 0.05$) with obesity (**Fig 2**). Interestingly, all genetic markers with higher MAF in obese study group also showed statistically significant and positive association with obesity (**Fig 2**). Of these markers, highly significant associations were observed in case of rs1137101, rs1558902, and rs2241766 genetic variants of *LEPR*, *FTO*, and *ADIPOQ* genes (OR = 2.50–4.42). Whereas three *FTO* genes SNPs rs9939609, rs1421085 and rs17817449 produced statistically significant yet mild association (OR = 1.03–1.44) towards raised BMI. The BMI raising effect size analysis per risk allele copy of above studied variants also confirmed positive correlation of the minor alleles of rs1137101, rs1558902 and rs2241766 SNPs with obesity risk (**Fig 3**). The only exception is rs7204609, rs3764261 and rs7799039 SNPs of *FTO*, *CRPT* and *LEP* genes, the negative effect sizes (β = -0.155, -0.125, -0.103Kg/m$^2$) indicate BMI lowering, obesity protective effect of their minor allele.

The BMI raising effect size of the minor G allele of rs1137101 variant was second highest in magnitude in total studied population (β = 0.285 Kg/m$^2$) but highest among all studied variants when compared between male and female subjects (β = 0.281–0.294 Kg/m$^2$, p = 7.5x10$^{-20}$). Thus, our results confirm that the G allele of rs1137101*LEPR* gene SNP stands out to be an informative obesity risk variant, increasing the susceptibility of general Pakistani populations' minor allele carriers towards weight gain. *LEPR* receptor in interactions with *LEP* gene has been reported for strong obesogenic influences in some world ethnicities while others are contradictory [43–47]. However, the G allele of rs7799039 *LEP* SNP showed protective association with over-weight-obese BMI and a statistically significant BMI lowering effect of G allele (β = -0.103 Kg/m$^2$) in studied Pakistani cohort.

*CETP* gene variant rs3764261 is not well studied for its association with BMI. Based on its strong association with lean BMI as well as strong BMI lowering effect in our studied Pakistani population clearly indicates its possible role in maintaining healthy body weight. The estimated effect sizes of BMI raising risk alleles of all genotyped SNPs are reported in (**Fig 3**).

The rs2241766 SNP of *ADIPOQ* gene showed statistically significant association with BMI. For the BMI increasing effect size estimation, the minor G allele of rs2241766 SNP maintained

high effect value ($\beta$ = 0.239Kg/m$^2$).The *ADIPOQ* gene encodes hormone adiponectin synthesized by adipocytes which helps in the metabolism of fats and carbohydrates whereby its low levels have been reported significantly correlated with elevated BMI [48–50]. The rs2241766 SNP of *ADIPOQ* is thought to be a candidate marker for obesity due to its significant association with BMI-defined obesity in various populations [51–54]. However, some contrasting reports have been linked with regional differences in frequency of *ADIPOQ* polymorphism [48].

The *FTO* gene has been shown to be strongly associated with obesity against the increase in the food intake and BMI [16, 54]. Genome-wide association studies have identified associations of SNPs in the first intron of *FTO* with BMI and obesity that have been replicated in different populations and age groups [55–57]. Among *FTO* SNPs, the rs17817449 and rs9939609 are extensively studied markers. Strong association of rs17817449 with obesity/raised BMI has been reported in European, Korean, Egyptian and North Indian cohorts, whereas African Americans lacked its association with obesity [58–63]. The rs9939609 SNP is among widely screened genetic variants in different population cohorts worldwide and is considered as a candidate marker for obesity screening [64–67]. This variant has attracted particular interest of obesity researchers because of its strongest BMI increasing effect [68, 69]. The rs1421085 and rs1558902 SNPs have also been reported to be strongly associated with anthropometric traits especially BMI, while excessive eating behaviors have been reported for association with rs1558902 SNP [70–72]. With regards to their risk alleles BMI raising effect size estimation, rs1558902 variant showed larger effect ($\beta$ = 0.26 Kg/m$^2$, p = 1x10$^{-6}$) as compared to rs17817449 ($\beta$ = 0.249 Kg/m$^2$, p = 1x10$^{-5}$). Interestingly, thers9941349 *FTO* SNP had a negative effect ($\beta$ = -0.125 Kg/m$^2$, p = 1x10$^{-4}$), which clearly indicates its significant protective effect from excess weight gain/obesity perhaps via maintenance of healthy lifestyle. Our findings showing a BMI lowering, protective effect of SNP rs9941349 of *FTO* gene are contrasting with previous studies conducted on African derived population which showed correlation of polymorphism with extreme obesity [73, 74].

The only genetic variant which not only showed protective association towards obesity but significant positive association with lean body mass was rs3764261 SNP of *CETP* gene. The BMI effect size of rs3764261 variant resulted in a significant decrease in BMI ($\beta$ = -0.155 Kg/m$^2$) per copy of its minor/protective A allele. The *CETP* gene is mostly reported for its association with lipid profile especially elevated HDL levels [75]. However, the association of rs3764261 variant with BMI is not well studied. Based on its strong association with lean BMI along with a strong BMI lowering effect in Pakistani individuals clearly indicate its protective allele carriers are more susceptible towards lean body mass. The estimated effect sizes of BMI raising risk alleles of all genotyped SNPs are reported in **Fig 3**.

## Conclusions

Findings of present study confirm that common forms of human obesity are polygenic and could be multifactorial involving interactions among multiple genetic markers and obesogenic environmental factors [37]. The BMI increasing alleles of FTO, LEPR and ADIPOQ genetic variants are common in Pakistani individuals thus increasing their risk towards obesity. It is worth mentioning the obesity protective genetic variants especially CEPT which promotes lean body mass in Pakistani individuals carrying its minor allele. However, there is need for replicating our findings by conducting future association studies in larger cohorts representing different ethnic groups and genetic-environmental interactions. This may help computing the individual genetic risk scores to assess the risk/protective effect of a set of obesity candidate genetic markers and may aid in the development of population-based obesity screening and prevention strategies.

## Supporting information

**S1 File.**
(DOCX)

## Author Contributions

**Conceptualization:** Muhammad Saqlain, Muhammad Nawaz, Ghazala Kaukab Raja.

**Data curation:** Muhammad Saqlain, Madiha Khalid, Muhammad Fiaz, Sadia Saeed, Asad Mehmood Raja, Muhammad Mobeen Zafar, Tahzeeb Fatima, Muhammad Nawaz, Ghazala Kaukab Raja.

**Formal analysis:** Muhammad Saqlain, Muhammad Fiaz, Sadia Saeed, Muhammad Mobeen Zafar, Tahzeeb Fatima, João Bosco Pesquero, Cristina Maglio, Hadi Valadi, Muhammad Nawaz, Ghazala Kaukab Raja.

**Funding acquisition:** Ghazala Kaukab Raja.

**Investigation:** Sadia Saeed, Asad Mehmood Raja, Tahzeeb Fatima, João Bosco Pesquero, Cristina Maglio, Hadi Valadi, Ghazala Kaukab Raja.

**Methodology:** Muhammad Saqlain, Madiha Khalid, Asad Mehmood Raja, Muhammad Mobeen Zafar.

**Project administration:** Ghazala Kaukab Raja.

**Resources:** Muhammad Nawaz, Ghazala Kaukab Raja.

**Software:** Madiha Khalid, Tahzeeb Fatima, Ghazala Kaukab Raja.

**Supervision:** Muhammad Nawaz, Ghazala Kaukab Raja.

**Validation:** Sadia Saeed, João Bosco Pesquero, Cristina Maglio, Hadi Valadi, Muhammad Nawaz, Ghazala Kaukab Raja.

**Visualization:** Ghazala Kaukab Raja.

**Writing – original draft:** Muhammad Saqlain, Ghazala Kaukab Raja.

**Writing – review & editing:** Muhammad Saqlain, Madiha Khalid, Muhammad Fiaz, Sadia Saeed, Asad Mehmood Raja, Muhammad Mobeen Zafar, Tahzeeb Fatima, João Bosco Pesquero, Cristina Maglio, Hadi Valadi, Muhammad Nawaz, Ghazala Kaukab Raja.

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
