## [Decision Letter · Decision Letter 0]

31 May 2022

PONE-D-22-10727Risk variants of obesity associated genes demonstrate BMI raising effect in a large cohortPLOS ONE

Dear Dr. Nawaz,

Thank you for submitting your manuscript to PLOS ONE. After careful consideration, we feel that it has merit but does not fully meet PLOS ONE’s publication criteria as it currently stands. Therefore, we invite you to submit a revised version of the manuscript that addresses the points raised during the review process.

We look forward to receiving your revised manuscript.

Kind regards,

Nidaa Ababneh

Academic Editor

PLOS ONE

Journal Requirements:

Reviewers' comments:

Reviewer's Responses to Questions

**Comments to the Author**

1. Is the manuscript technically sound, and do the data support the conclusions?

Reviewer #1: Yes

Reviewer #2: Partly

Reviewer #3: Yes

2. Has the statistical analysis been performed appropriately and rigorously? 

Reviewer #1: Yes

Reviewer #2: Yes

Reviewer #3: I Don't Know

3. Have the authors made all data underlying the findings in their manuscript fully available?

Reviewer #1: Yes

Reviewer #2: No

Reviewer #3: Yes

4. Is the manuscript presented in an intelligible fashion and written in standard English?

Reviewer #1: Yes

Reviewer #2: No

Reviewer #3: Yes

5. Review Comments to the Author

Reviewer #1: Obesity, a rapidly growing health risk worldwide, is a polygenic disease associated with multiple genetic variants. Here, Saqlain and colleagues assessed if single nucleotide polymorphisms (SNPs) in adiponectin, C1Q and collagen domain containing (ADIPOQ), cholesteryl ester transfer protein (CEPT), FTO alpha-ketoglutarate dependent dioxygenase (FTO), leptin (LEP), and leptin receptor (LEPR) genes are associated with obesity in 4000 subjects of mixed ethnicities stratified by BMI. In the obese BMI group, six SNPs with higher MAFs were significantly associated with strong risk effect. In contrast, three variants had higher MAFs in the lean-normal BMI group and were protective in over-weight-obese group. Only one SNP was associated with the lean BMI. In summary, Saqlain and colleagues identified SNPs in ADIPOQ, FTO and LEPR genes that were associated with obesity in a mixed ethnicity population.

General Comments

This is an interesting study that examined the association between candidate SNPs and obesity stratified by BMI in a mixed ethnicity population. Overall, the manuscript is well-written, confirm polygenic nature of the common forms of obesity and highlight the complexity across ethnicities. Addressing the following specific comments will enable the reader to interpret the results and gain insight into the potential implications of the findings.

Specific Comments

1. The authors indicate that the study cohort was of mixed ethnicity, but it is unclear if the MAFs were ethnic-specific.

2. While the findings are intriguing and a genetic association study in a Pakistani cohort is novel, the association between candidate SNPs and BMI has previously been reported.

3. Discussing the potential mechanisms by which the identified SNPs are associated with BMI may provide important insights for the reader.

4. As the investigators acknowledge, replicating their findings in an independent Pakistani cohort will be important especially for risk stratification.

Reviewer #2: My main concern is whether the authors verified or repeated genotyping because the allelic frequency obtained in most of the studied SNPs shows a significant difference in relation to the global and SAS MAFs (Table 2). Practically all the SNPs studied in Pakistani showed MAF very different from those described in the reported populations. A reason which make think about if genotyping was correctly done. Authors are invited to describe in the materials and methods section how many samples or what percentage of samples were repeated to verify correct genotyping.

In addition to these notable differences between all the MAFs, it is also noteworthy that for the SNP rs1558902 of the FTO gene it reports 0 MAF in the normal weight category, while in the range of categories from lean to obese it is reported a range from 0.39 to 0.63 (figure 1), other words, frequency of allele A of rs 1558902 is completely absent in normal subjects wheras lean, overwiight and obese subject had MAFs ranging 0.39 - 0.64.

Moreover they reported for LEP rs7799039 a MAF of 0.66 in table 1, whereas in figure 1 ranging of this SNP is from 0.25 to 0.43 for all categories, which sounds an inconsistency.

In introducction, authors are suggested to support the metabolic or phisiological basis to select those genetic variants. Explain exactly which is the Impact of the risk allele in order to support the Association of those SNPs.

Exsclussion criteria mentioned that subjetcs with comorbilities or some illness were excluded. Subjects also seems to be adults since mínimum age was 20 years in table 1, however anthropometric parametes such as mínimum Weight of 19 kgrs and minimun height of 95 cm are not consistent with adult parameters, even being under-weight.

The manuscript reports anthropometric parameters such WC and metabolic parameters such as HDL, LDL, etc. that were not analyzed in association to genotype. Discussion do not talk about correlación of genética variants and mean antrhopometric or metabolic parameters, Or according to genotype.

Authors are suggested to explain figures 2 an 3 or organized information of these figures in tables in order to clarification about the significant results which clearly shows significant statistics such as p valuss and interval of confidence (IC) other than OR, as well as to report sample size or N of each category, and proportion of sexes (figure 3). It is difficult for a reader to replicate statistical analysis without data such as N of each category and results of p and IC

Correct the acronym for BSF in table 1

The manuscript needs proofreading

Reviewer #3: The manuscript is very interesting because the authors report identification of BMI raising risk markers, risk/MAFs, independent associations with BMI based body mass categories, and BMI raising risk effect. Beside that, Pakistani population lacks prior studies on genetic susceptibility to weight gain and genotype distribution data of potential obesity predisposing genetic variants.

The study is well described, was appreciated by a research ethics committee, and has a very interesting number of subjects included. In addition, the authors present an overview of different SNPs and their genes that affect the population included in the study. Statistical analyzes are well described and presented in graphs and figures. As a weakness of the study, presented by the authors themselves, is the lack of data related to lifestyle. It is important to highlight still that the authors mention in the methodology, but do not cite results or discuss: waist circumference, blood pressure (SBP and DBP), fasting blood sugar, triglycerides, total cholesterol, high density lipoprotein cholesterol, low density lipoprotein cholesterol, and very low density lipoprotein cholesterol. In table 1, it is questioned whether the upper limit of the BMI is correct (BMI 62.48kg/m2).

6. PLOS authors have the option to publish the peer review history of their article (what does this mean?). If published, this will include your full peer review and any attached files.

Reviewer #1: **Yes: **Dawood DARBAR

Reviewer #2: No

Reviewer #3: **Yes: **Andréia Rosane de Moura Valim

---

## [Author Response · Author response to Decision Letter 0]

21 Aug 2022

Dear Dr., Nidaa Ababneh

Academic Editor, PLOS ONE, 

Thank you for giving us the chance to revise our manuscript (#PONE-D-22-10727) in the light of reviewers’ comments, which have been very helpful for improving our manuscript. We are grateful for their time, the careful review and constructive comments made by referees. We have revised our manuscript and added the changes asked/suggested by reviewers, which can be seen with tracked changes in the revised file. 

Comments from editorial office 

The recommended style has been applied in the revised version. 

The funding information has been updated. 

Answer: The recommended reference style has been updated. 

Additional comments from editorial office, 

1. Please note that funding information should not appear in the Acknowledgments section or other areas of your manuscript. We will only publish funding information present in the Funding Statement section of the online submission form. Please remove any funding-related text from the manuscript. 

Answer: The funding information is now removed from the text, and is available online only. 

Answer: The ethics statement appears only the methods. It was deleted from other places from the text

3. Data availability statement: 

Answer: Data statement is updated. The following statement is added now. ‘’All data generated or analyzed during this study are included within this article’’. Additionally, source data is included in supplementary information. 

Review Comments to the Author 

The point-by-point response to each reviewer is provided below.

Reviewer #1: 

Obesity, a rapidly growing health risk worldwide, is a polygenic disease associated with multiple genetic variants. Here, Saqlain and colleagues assessed if single nucleotide polymorphisms (SNPs) in adiponectin, C1Q and collagen domain containing (ADIPOQ), cholesteryl ester transfer protein (CEPT), FTO alpha-ketoglutarate dependent dioxygenase (FTO), leptin (LEP), and leptin receptor (LEPR) genes are associated with obesity in 4000 subjects of mixed ethnicities stratified by BMI. In the obese BMI group, six SNPs with higher MAFs were significantly associated with strong risk effect. In contrast, three variants had higher MAFs in the lean-normal BMI group and were protective in over-weight-obese group. Only one SNP was associated with the lean BMI. In summary, Saqlain and colleagues identified SNPs in ADIPOQ, FTO and LEPR genes that were associated with obesity in a mixed ethnicity population.

General Comments

This is an interesting study that examined the association between candidate SNPs and obesity stratified by BMI in a mixed ethnicity population. Overall, the manuscript is well-written, confirm polygenic nature of the common forms of obesity and highlight the complexity across ethnicities. Addressing the following specific comments will enable the reader to interpret the results and gain insight into the potential implications of the findings.

Specific Comments

1. The authors indicate that the study cohort was of mixed ethnicity, but it is unclear if the MAFs were ethnic-specific. 

Answer: 

We thank the reviewer for pointing out this. The MAFs were not ethnic-specific. We have updated the text. 

2. While the findings are intriguing and a genetic association study in a Pakistani cohort is novel, the association between candidate SNPs and BMI has previously been reported. 

Answer: 

Thank you for the suggestion, we have provided relevant references for this, and have discussed this in the discussion. 

3. Discussing the potential mechanisms by which the identified SNPs are associated with BMI may provide important insights for the reader. 

Answer: 

Thank you for the suggestion. We have included more information about this, however true mechanisms for every SNP are not fully understood. 

4. As the investigators acknowledge, replicating their findings in an independent Pakistani cohort will be important especially for risk stratification. 

Answer: 

We agree with the reviewer, and have emphasized the need of replicating these finding in future studies. 

Reviewer #2: 

1. My main concern is whether the authors verified or repeated genotyping because the allelic frequency obtained in most of the studied SNPs shows a significant difference in relation to the global and SAS MAFs (Table 2). Practically all the SNPs studied in Pakistani showed MAF very different from those described in the reported populations. A reason which makes think about if genotyping was correctly done. Authors are invited to describe in the materials and methods section how many samples or what percentage of samples were repeated to verify correct genotyping. 

Answer: 

We thank the reviewer for highlighting this deficiency. These MAFs are not unexpected. Actually 50% individuals of our population cohort consisted of over-weight and obese and another 25% lean, therefore we could not expect MAFs of similar ranges as expected in healthy population groups. We use to repeat 10-12% samples. 

2. In addition to these notable differences between all the MAFs, it is also noteworthy that for the SNP rs1558902 of the FTO gene it reports 0 MAF in the normal weight category, while in the range of categories from lean to obese it is reported a range from 0.39 to 0.63 (figure 1), other words, frequency of allele A of rs1558902 is completely absent in normal subjects whereas lean, overweight and obese subject had MAFs ranging 0.39 - 0.64. 

Answer: 

We apologize for this discrepancy. It was typing error, which has been corrected now. 

3. Moreover, they reported for LEP rs7799039 a MAF of 0.66 in table 1, whereas in figure 1 ranging of this SNP is from 0.25 to 0.43 for all categories, which sounds an inconsistency. 

Answer: 

Table 2 presents globally accepted major and minor alleles of all genetic markers. However, in Figure 1 the G allele of LEP rs7799039 SNP was considered as minor due to very high frequency of its A allele in current population cohort. All other analysis were also performed using G as minor allele. A statement is included in the results section. 

4. In introduction, authors are suggested to support the metabolic or physiological basis to select those genetic variants. Explain exactly which is the Impact of the risk allele in order to support the Association of those SNPs. 

Answer: 

Thank you for this suggestion. We have explained the Impact of the risk allele in order to support the Association of those SNPs. 

5. Exclusion criteria mentioned that subjects with comorbidities or some illnesses were excluded. Subjects also seems to be adults since minimum age was 20 years in table 1, however anthropometric parameters such as minimum Weight of 19 kg and minimum height of 95 cm are not consistent with adult parameters, even being under-weight. 

Answer: 

We have corrected the exclusion criteria, and have made the minimum age, weight and height consistent with adult parameters. Information has been updated in the main manuscript 

6. The manuscript reports anthropometric parameters such WC and metabolic parameters such as HDL, LDL, etc. that were not analyzed in association to genotype. Discussion do not talk about correlation of genetic variants and mean anthropometric or metabolic parameters, or according to genotype. 

Answer: The manuscript included descriptive statistics of all relevant anthropometric and clinical variables in overall study population. However, keeping with main objective of our study, only BMI was included in the genetic association and BMI raising risk allele effect analysis. 

7. Authors are suggested to explain figures 2 and 3 or organized information of these figures in tables in order to clarification about the significant results which clearly shows significant statistics such as p values and interval of confidence (IC) other than OR, as well as to report sample size or N of each category, and proportion of sexes (figure 3). It is difficult for a reader to replicate statistical analysis without data such as N of each category and results of p and IC. 

Answer: 

Thank for this suggestion. While keeping the figures instead of tables, we have updated the figures with better presentation, and explanation what reviewer has pointed out. We have provided the source data/tables which support figures 2 and 3 with complete analysis like OR, CI, beta-values and p-values. The number of individuals for each BMI group and male female numbers also added in the source data tables. 

8. Correct the acronym for BSF in table 1. 

Answer: 

BSF was corrected to FBS. 

9. The manuscript needs proofreading 

Answer: 

We have carefully read the manuscript and have made the corrections.

Reviewer #3: 

The manuscript is very interesting because the authors report identification of BMI raising risk markers, risk/MAFs, independent associations with BMI based body mass categories, and BMI raising risk effect. Besides that, Pakistani population lacks prior studies on genetic susceptibility to weight gain and genotype distribution data of potential obesity predisposing genetic variants.

The study is well described, was appreciated by a research ethics committee, and has a very interesting number of subjects included. In addition, the authors present an overview of different SNPs and their genes that affect the population included in the study. Statistical analyzes are well described and presented in graphs and figures. 

1. As a weakness of the study, presented by the authors themselves, is the lack of data related to lifestyle. It is important to highlight still that the authors mention in the methodology, but do not cite results or discuss: waist circumference, blood pressure (SBP and DBP), fasting blood sugar, triglycerides, total cholesterol, high density lipoprotein cholesterol, low density lipoprotein cholesterol, and very low density lipoprotein cholesterol. In table 1, it is questioned whether the upper limit of the BMI is correct (BMI 62.48kg/m2). 

Answer: 

We are thankful to reviewer for the comments on our manuscript. We have taken this into consideration and have updated the description. 

Sincerely, 

Dr. Ghazala Raja, and Dr. Muhammad Nawaz

---

## [Editor Report · Decision Letter 1]

7 Sep 2022

Risk variants of obesity associated genes demonstrate BMI raising effect in a large cohort

PONE-D-22-10727R1

Dear Dr. Nawaz,

We’re pleased to inform you that your manuscript has been judged scientifically suitable for publication and will be formally accepted for publication once it meets all outstanding technical requirements.

Kind regards,

Nidaa Ababneh

Academic Editor

PLOS ONE

---

## [Editor Report · Acceptance letter]

11 Sep 2022

PONE-D-22-10727R1 

Risk variants of obesity associated genes demonstrate BMI raising effect in a large cohort 

Dear Dr. Nawaz:

I'm pleased to inform you that your manuscript has been deemed suitable for publication in PLOS ONE. Congratulations! Your manuscript is now with our production department. 

Kind regards, 

on behalf of

Dr. Nidaa Ababneh 

Academic Editor

PLOS ONE